# Predictive Factors for Local Recurrence after Intraoperative Microwave Ablation for Colorectal Liver Metastases

**DOI:** 10.3390/cancers15010122

**Published:** 2022-12-25

**Authors:** Yoshiyuki Wada, Yuko Takami, Tomoki Ryu, Yoshihiro Uchino, Tota Kugiyama, Yoriko Nomura, Hideki Saitsu

**Affiliations:** Department of Hepato-Biliary-Pancreatic Surgery, Clinical Research Institute, National Hospital Organization Kyushu Medical Center, Fukuoka 810-8563, Japan

**Keywords:** microwave, ablation, local recurrence, liver metastases, predictive factors

## Abstract

**Simple Summary:**

The European Society of Medical Oncology and National Comprehensive Cancer Network guidelines indicate that ablation is recommended as a stand-alone treatment or in combination with resection for colorectal liver metastases (CRLM) as long as all visible tumors are eradicated. However, local recurrence (LR) is a major setback for microwave ablation (MWA). Even though several LR predictors have been reported, those following intraoperative MWA still remain unclear. Therefore, we retrospectively analyzed CRLM cases to determine LR predictive factors following intraoperative MWA. The minimal ablation margins were measured using anatomic landmarks, and a multivariate analysis for LR predictors after intraoperative MWA revealed that the ablation margin was the most powerful predictor per patient and lesion. In addition, anatomical location (i.e., the posterosuperior segment defined as segments 1, 7, and 8), tumor size >15 mm, tumor size > 20 mm, and proximity to Glisson were the independent LR predictors per lesion.

**Abstract:**

This study aimed to clarify local recurrence (LR) predictive factors following intraoperative microwave ablation (MWA) for colorectal liver metastases. The data from 195 patients with 1392 CRLM lesions, who were preoperatively diagnosed by gadolinium-enhanced MRI with diffusion-weighted imaging and dynamic CT and treated with intraoperative MWA (2450 MHz) with or without hepatectomy, from January 2005 to December 2019, were retrospectively reviewed and analyzed using logistic regression. In addition, the margins were measured on contrast-enhanced CT 6 weeks post-ablation. Overall, 1066 lesions were ablated. The LRs occurred in 44 lesions (4.1%) among 39 patients (20.0%). The multivariate analysis per patient showed that tumor size > 20 mm and ablation margin < 5 mm were significant predictors for LR. Furthermore, multivariate analysis per lesion revealed that segments 1, 7, and 8 and tumor size > 15 mm, ablation margin < 5 mm, tumor size > 20 mm, and proximity to the Glisson were significant LR predictors. Finally, the outcome of this study may help determine indications for MWA.

## 1. Introduction

The third most common cancer worldwide is colorectal cancer (CRC). Approximately 25–30% of patients present with synchronous metastases at the time of diagnosis, and a further 40% develop metachronous metastases during the course of the disease [1]. Additionally, approximately 50% of patients with CRC develop liver metastases [2]. Resection is the most effective treatment for colorectal liver metastases (CRLM) [3].

Non-surgical alternative treatments have been introduced for patients with unresectable CRLM. Among these, radiofrequency ablation (RFA) is the most frequently used. In a randomized phase II study (the CLOCC trial) on patients with unresectable CRLM, RFA significantly affected progression-free and overall survival [4]. In addition, RFA is widely used in the surgical and radiological communities as a reliable treatment modality for patients with unresectable CRLM [5].

Furthermore, in recent times, the National Comprehensive Cancer Network (NCCN) [6] and the European Society of Medical Oncology (ESMO) [7] guidelines indicate that ablation is recommended as a stand-alone treatment or in combination with hepatic resection (HR) for CRLM as long as all visible tumors are eradicated.

RFA and microwave ablation (MWA) are commonly employed for the local treatment of malignant liver tumors, including CRLM. Additionally, local hyperthermia causes death via coagulation necrosis in both thermal ablation techniques [8,9]. Further, MWA has several advantages over RFA, including higher intratumoral temperatures, larger ablation zones in high-perfusion areas, shorter ablation times, and decreased susceptibility to the heat-sink effect [10,11].

However, local tumor recurrence is a major setback associated with ablation therapy, which occurs in MWA and RFA. Local treatment failure ranging from 20 to 60% has been reported with RFA [12,13] and 6 to 20% with MWA [11,14,15]. The local recurrence (LR) rate may be lower for patients with CRLM treated with MWA than those treated with RFA [14]. On the other hand, it was reported that local control did not significantly differ between RFA and HR in selected patients [16].

The tumor size [9,11,12,15,17,18,19,20], tumor location (subcapsular [9] and perivascular [11,13,17]), and ablation margin size [18,20,21] are LR prognostic factors for ablation. Recently, ablation margin size has been considered the most important factor for LR after ablation [20,21,22]. Because MWA reduces the heat sink effect, it was considered that perivascular locations were unrelated to MWA local recurrence [21]. However, this is still a controversial issue [17]. The ablation approach (percutaneous, open, or laparoscopic) did not affect the local control of ablation [23]. However, the relationship between the ablation margin and LR after surgical MWA remains unclear.

Therefore, this study aimed to retrospectively review patients with CRLM treated with surgical MWA in order to identify the predictive factors for LR after intraoperative MWA for CRLM and investigate the relationship between ablation margin and LR.

## 2. Materials and Methods

### 2.1. Patients

The data were retrospectively analyzed for patients who underwent surgical MWA with or without hepatic resection (HR) for initial CRLM at our institution. In the data collected from January 2005 to December 2019, 284 consecutive patients with initial CRLM underwent surgical treatment (MWA, HR, or MWA + HR). Among these, 199 patients underwent MWA with or without HR.

Prior to surgical treatment, each patient was informed of the treatment options for CRLM, following Japanese guidelines. However, even if HR was optimal, MWA or MWA + HR was allowed when patients were unfit for resection (i.e., older adults with low functional status or multiple comorbidities) or the patient desired and selected MWA or MWA + HR. As a result, a multidisciplinary colon cancer or hepatobiliary disease management team made the decision to perform ablation rather than resection or radiotherapy on an individual basis to provide the best possible treatment for each patient.

The inclusion criteria were as follows: (1) CRLM with no previous locoregional treatment by ablation, embolization, or resection; (2) all target CRLM lesions with assessment by contrast-enhanced computed tomography (dynamic CT) or gadolinium-enhanced magnetic resonance imaging (MRI) preoperatively; (3) histopathological diagnosis of CRLM by intraoperative biopsy in at least one or more lesions; (4) CT evidence of effective ablation: According to the guidelines [24] for evaluating treatment efficacy, all ablations would have been essentially assessed 6 weeks after MWA contrast-enhanced CT as the first post-assessment. An ablation defect completely covering the target CRLM was considered a complete and technically effective MWA.

The two patients who dropped out during the follow-up period within 2 years and the two who had no radiological evaluation of treatment efficacy within 6 weeks were excluded. Overall, 195 patients with CRLM were enrolled. The following data were available for curatively operated patients: patient characteristics, clinicopathological features, characteristics of hepatic lesions, date of recurrence, date of local recurrence, condition of local recurrence, date of death, or last visit.

Patients were followed up until they died or had their last visit. This study protocol was approved by the Institutional Review Board (No. 19C114) of Kyushu Medical Center and performed following the Declaration of Helsinki and the ethical guidelines for clinical studies in Japan. Furthermore, the requirement for written informed consent was waived owing to the retrospective nature of this study.

### 2.2. Microwave Ablation

The MWA was performed according to our standardized methods, as previously reported [16,17,18,19], using an open approach. An intraoperative ultrasound was routinely performed in all patients to identify lesions and monitor treatment effects and ablation margins. Furthermore, when ablation was used to treat small-sized CRLM, multiple CRLM, or CRLM lesions that were unclearly expressed by intraoperative ultrasonography, intraoperative contrast-enhanced ultrasonography was performed. Using a 2450-MHz system microwave generator (Alfre’s Pharma, Osaka, Japan), the ablation power settings and duration were a single coagulation at 40 W for 30 s per pulse using a 16-gauge, 150-mm-long needle, while 80–85 W for 30 s per pulse using a 21-gauge, shallow (range, 10–30 mm) could only be used for superficial CRLM tumors. A unique intraoperative MWA procedure involving electrode insertion and irradiation was repeated from the tumor surroundings to the center with multiple puncture points using the free-hand technique to avoid iatrogenic dissemination of tumor cells. An overlapping ablation area of 1 cm diameter per single session from the tumor surroundings to the inside formed a spherical ablation area with an intentional ablation margin of 5–10 mm.

### 2.3. Definitions

Gadolinium (Gd)-enhanced MRI with diffusion-weighted imaging (DWI) (from 2005 to 2007) or Gd-ethoxybenzyl-diethylenetriamine pentaacetic acid (Gd-EOB-DTPA) enhanced MRI with DWI and hepatobiliary phase imaging (from 2008) were used to preoperatively diagnose all targeted CRLM. In addition, dynamic CT scans with a 1 mm slice were also preoperatively examined for the diagnosis of CRLM, and all targeted CRLM were reconfirmed as solid tumors by intraoperative ultrasonography.

The tumor size was determined using intraoperative ultrasonography.

The absence of a residual tumor within 1 cm of the ablation site in the first control imaging with contrast-enhanced CT 6 weeks after MWA was defined as complete ablation of a CRLM lesion.

Furthermore, local recurrence was defined as tumor recurrence at the site of ablation after at least one imaging without evidence of a resting tumor. The intrahepatic recurrence was defined by the presence of a tumor in any other liver site. Finally, the recurrence was defined as local recurrence, intrahepatic recurrence, or extrahepatic recurrence.

The clinical risk score was defined by a disease-free interval from primary to liver metastasis ≤ 12 months, more than one liver tumor, the largest hepatic metastasis ≥ 3 cm, carcinoembryonic antigen (CEA) level of > 30 ng/mL, and a node-positive primary [25].

The couinaud’s segments 1, 7, and 8 were defined as the posterosuperior segments [15].

Contact with the hepatic vein (HV)/ inferior vena cava (IVC) was defined as present when a lesion was in direct contact with the first or second trunk of the HV or IVC. The proximity to the Glissonian pedicle was defined as present when a lesion was located <5 mm of the first to the third branch of the Glissonian pedicle.

The ablation margins were measured using the last 6 weeks of contrast-enhanced CT scans and anatomic landmarks, as described by Wang et al. [22]. The distances were measured from the edge of the tumor/ablation defect to the anatomic landmarks chosen on the pre- and postoperative contrast-enhanced CT scans, respectively [21,22]. In addition, for each landmark, the pre-ablation distance was subtracted from the post-ablation distance to give the margin at that site; the smallest value was considered the minimal margin. The margin size was classified as: 0 mm, 1 to <5 mm, 5 to <10 mm, and 10 mm or more. Further, although lesions were detected by intraoperative ultrasonography and the first preoperative Gd-enhanced MRI with DWI and contrast-enhanced CT scans before preoperative chemotherapy, when an unclear lesion showed at the last preoperative Gd-enhanced MRI with DWI or contrast-enhanced CT scans after preoperative chemotherapy, the minimal margin was regarded as not being evaluated.

### 2.4. Follow Up

The total number of patients were regularly screened for recurrence by monitoring the plasma levels of CEA and carbohydrate antigen (CA) 19-9 every 2–3 months, ultrasonography every 2–3 months, and dynamic CT or contrast-enhanced MRI every 4–6 months.

Additionally, when the database was locked (in February 2022), the median follow-up time was 46.3 months.

### 2.5. Statistical Analysis

The medians and ranges of continuous data were compared using the Mann–Whitney U-test. Categorical data were compared using the Pearson chi-square or Fisher’s exact test, as appropriate. The statistical significance level was set at *p* < 0.05.

The logistic regression analysis was used to identify the independent predictive factors for local recurrence. Moreover, variables with a value < 0.10 in the univariate analysis were included in the multivariate logistic regression analysis. The statistical significance level was set at *p* < 0.05.

The overall survival (OS) was defined as the interval from MWA to death or the date of the last or most recent follow-up visit. The disease-free survival (DFS) was defined as the interval from MWA to the date of diagnosis of the first recurrence or last follow-up. The local tumor’s progression-free survival (LTPFS) was defined as the interval from MWA to the date of diagnosis of the local recurrence or last follow-up.

Finally, multivariate analysis was performed using a Cox proportional hazards model to identify potential covariates associated with these prognostic indicators. Factors identified as showing at least marginal association with prognostic indicators (*p* < 0.05) in the univariate analysis were entered as covariates in a multivariate Cox proportional hazard analysis model of the backward elimination procedure. The statistical significance level was set at *p* < 0.05.

The statistical analyses were performed using the JMP version 14.0 software (SAS Institute Inc., Cary, NC, USA).

## 3. Results

Between 2005 and 2019, 195 CRLM patients with 1392 CRLM tumorous lesions underwent surgical treatment at our institution. In addition, MWA ablation was performed for 1066 CRLM lesions and concomitant hepatectomy for 326 CRLM lesions. While preoperative radiological examination confirmed that all targeted CRLM showed a hypovascular tumor with ring enhancement by Gd-enhanced MRI and dynamic CT, high intensity with T2-weighted imaging and Gd-enhanced MRI with DWI, and low intensity with the hepatobiliary phase image by Gd-EOB-DTPA enhanced MRI (from 2008). Furthermore, all targeted CRLMs were confirmed as solid tumors by intraoperative ultrasonography. Finally, 411 lesions (38.6%) were histopathologically diagnosed as CRLM by intraoperative biopsy after MWA. All patients were followed for more than two years. The median survival time was 5.0 years. OS at 1, 3, and 5 years was 92.8%, 67.5%, and 49.9%, respectively (Figure 1). DFS were 39.1%, 23.9%, and 21.7% at 1, 2, and 3 years, respectively (Figure 2).

In this study, the complete ablation was achieved in 100% of CRLM lesions (1066/1066) by 6 weeks after contrast-enhanced CT estimation.

The local recurrence after MWA was detected in 39 patients (20.0%) within 2 years after MWA. Figure 3 shows the LTPFS.

The LTPFS were 83.7% and 80.2% at 1 and 2 years, respectively. The maximum period of local recurrence was 729 days after surgery. Compared with the patients’ characteristics between the local recurrence and no local recurrence groups, the timing of CRLM, the largest diameter of CRLM treated with MWA, the number of CRLM treated with MWA, and preoperative chemotherapy were significantly different (Table 1).

In Table 2, predictive factors for local recurrence following MWA treatment in CRLM are detailed. The regimen of preoperative chemotherapy in the local recurrence and no local recurrence groups was oxaliplatin-based doublet in four and four patients, irinotecan-based doublet in 0 and one, cisplatin-based doublet in three and three, bevacizumab and oxaliplatin-based doublet in eight and 27, bevacizumab and irinotecan-based doublet in two and three, epidermal growth factor receptor inhibitor and irinotecan-based doublet in four and eight, and bevacizumab and triplet in 0 and five patients, respectively (*p* = 0.33). In the univariate analysis, the timing of CRLM, the largest diameter of MRLM treated with MWA, the number of CRLM treated with MWA, preoperative chemotherapy, adjuvant chemotherapy, and minimal ablation margin (Figure 4) were the significant predictive factors for patients with local recurrence. In the multivariate analysis, the largest diameter > 20 mm of CRLM treated with MWA (odds ratio (OR), 4.79; 95% confidence interval (CI), 1.83–12.50, *p* < 0.01) and minimal ablation margin < 5 mm (OR, 8.32; 95% CI, 2.99–23.13, *p* < 0.01) were independent predictors for patients with local recurrence.

In focusing on lesions treated with MWA, 1066 CRLM tumorous lesions were ablated with MWA in all 195 patients. Of the 1066 lesions, local recurrence was detected in 44 (4.1%), with a median diameter of 23.0 (10–44.1) mm.

The characteristics of lesions treated with MWA are shown in Table 3. The location of Couinaud’s segment and positioning to the trunk of the HV, IVC, or segmental branch to the main trunk of the Glissonian pedicles were considered. Minimal ablation margin (0 mm, 0.1–5 mm, >5 to 10 mm, and >10 mm) was also considered. Local recurrences in segments 1, 7, and 8 were detected in segments 2, 22, and 12 (4.5, 13.6, and 5.3%, respectively). The local recurrence rate in the other segments was <2%, although the local recurrence rate in the posterosuperior segment (segments 1, 7, and 8) was higher. Regarding ablation margin, local recurrence was developed in 42.9% (18/42), 30.8% (20/65), 0.6% (5/874), and 0% (0/66) for 0 mm, 0.1–5 mm, >5 to 10 mm, and >10 mm ablation margins, respectively. Of the 42 lesions in direct contact with the HV/IVC, 18 (42.9%) had local recurrence. Of the 15 lesions with proximity to the Glissonian pedicle (segmental branch to the main trunk), seven (46.7%) had local recurrence.

The predictors of CRLM lesions with local recurrence after MWA were investigated (Table 4). In the multivariate analysis, the posterosuperior segment (segments 1, 7, and 8) and size > 15 mm (OR: 14.33, 95% CI: 5.36–45.06, *p* < 0.01), ablation margin < 5 mm (OR: 45.76, 95% CI: 14.26–145.85, *p* < 0.01), and proximity to the Glissonian pedicle (OR: 9.08, 95% CI: 2.19–39.58, *p* < 0.01) were identified as independent predictors of CRLM lesions with local recurrence after MWA.

Furthermore, of the 44 lesions with local recurrence, only one (2.3%) had no predictive factor. In addition, twelve (27.9%), 28 (65.1%), and two (4.7%) lesions had one, two, and three predictive factors, respectively.

In comparison with all lesions with 0, one, two, and three predictive factors (791, 227, 44, and two lesions), local recurrence was observed in one (1.3%), 12 (5.3%), 28 (63.6%), and two (50%) lesions with 0, one, two, and three predictive factors, respectively.

The relationship between local recurrence and oncological outcomes was also investigated. The predictors of disease-free survival are shown in Table 5. In the multivariate analysis, the number of CRLMs, CA19-9 values, and primary T-stage were identified as independent predictors of disease-free survival. Additionally, local recurrence was significant in the univariate analysis; however, it was insignificant in the multivariate analysis. The predictors of overall survival are shown in Table 6. CEA and CA19-9 levels, primary T-stage, and primary N-status were independent predictors of overall survival.

## 4. Discussion

This study investigated the characteristics and predictive factors for the local recurrence of CRLM tumors treated with surgical MWA. To the best of our knowledge, this is the cohort study for the largest number of CRLM lesions evaluating local recurrence following treatment with MWA for CRLM.

A number of predictive factors for local recurrence after ablation therapy, including MWA or RFA, have been proposed in the literature, such as tumor type (hepatocellular carcinoma vs. CRLM) [11,12,13,14,15], larger tumor size [9,11,12,15,18,26], proximity to the liver vasculature [9,11,12,15,17], smaller ablation margins [3,11,13], location of the lesions [9], and type of ablation modality [21]. This study showed that proximity to large vessels, tumor size, and the number of tumors treated with MWA were significantly associated with local recurrence. The ablation margin is the most important factor for local recurrence after ablation treatment for CRLM [21]. This study investigated the prognostic factors, including the ablation margin, and revealed that the ablation margin was the independent predictive factor with the highest hazard ratio. A previous study [27] reported that the ablation margin > 10 mm offered the best tumor control. Further, in being consistent with the above-mentioned study, this study also showed no local recurrence in the lesions with an ablation margin > 10 mm. However, an ablation margin > 10 mm increased the risk of biliary complications after ablation in patients with hepatic arterial infusion chemotherapy. Therefore, the relationship between biliary complications after MWA and the ablation margin in patients with preoperative systemic chemotherapy needs to be clarified in the future.

In addition, this study showed that the tumor location was significantly associated with local recurrence. The posterosuperior segments (1, 7, and 8) and tumor size > 15 mm were independent predictive factors for local recurrence. Subcapsular [18] or perivascular location [9,12] have been reported as predictive factors for local recurrence; however, the anatomical segment has not been reported. Therefore, we investigated patients with CRLM treated with surgical (intraoperative) MWA but not percutaneous MWA. In surgical MWA, the MWA procedure may be limited by anatomical location.

In the consideration of the reason for local recurrence in the posterosuperior segments, segment 7 was located at a deep lesion from the surface of the liver for surgical ablation. Furthermore, the long puncture distance may make complete ablation difficult because of stray punctures and the difficulty of detailed evaluation of the puncture line or ablation margin by intraoperative ultrasonography. Additionally, Segment 8 was mostly located behind the abdominal wall during the laparotomy. In particular, the location of the tumor in the hepatic dome requires a shallow puncture angle and a long puncture distance for ablation, which may prevent adequate ablation. On the other hand, Segment 1 is deeply located and enclosed in the major vessels, which requires a long puncture distance or limits the puncture line for ablation. Furthermore, the risk of thermal injury to the bile duct often inhibits ablation with a margin in segment 1. Thus, all these factors may result in a greater likelihood of local recurrence in the posterosuperior segment. However, smaller lesions can reduce the risk of local recurrence when the tumor is located in the posterosuperior segment. In this study, the predictive factor for the posterosuperior segment was identified by univariate analysis but not by multivariate analysis. These findings indicate that smaller-sized lesions have a low risk for local recurrence even if the lesions are located in an anatomical position that requires MWA surgery. However, the condition of the lesion may also affect treatment effectiveness.

The outcome of this study revealed that the local recurrence rates with 0, one, and two predictive factors for local recurrence were 0.8, 14.3, and 54.8%, respectively. This result suggests that MWA for CRLM without these predictive factors may be satisfactory, whereas MWA for CRLM with these two predictive factors should be considered critically owing to an increased risk for local recurrence following treatment.

The proximity to the main trunk of the HV or IVC is a risk factor for local recurrence [9,11,12,13,15,17]. The local recurrence of CRLM with proximity to the HV or IVC is caused by the cooling effect or heat-sink effect for RFA [9,11]. As a result, the relationship between local recurrence and proximity to HV for MWA is controversial [17,21]. However, MWA is less susceptible to the heat-sink effect [21]. This study revealed that proximity to the HV is related to local recurrence by univariate analysis but not by multivariate analysis. Proximity to HV may not be significantly related to local recurrence in MWA. However, when lesions close to the HV are ablated, it is difficult to achieve a sufficient ablation margin and an effective puncture line. The study showed that a high local recurrence rate of 42.9% was detected for lesions with direct contact with HV. As a result, for lesions near the HV, a careful MWA procedure is required.

The prognostic factors for OS have been reported for CRLM ablation. Additionally, modified clinical risk scores for ablation [18,20,21] have been proposed to categorize the treatment effect for OS. This study also investigated prognostic factors, including modified clinical risk scores. Unfortunately, the modified clinical risk score was not significantly related to local recurrence, DFS, or OS. Since local recurrence was not significantly related to DFS or OS in this study, almost all patients exhibited multiple CRLM in the local recurrence group, and intrahepatic new recurrence in other liver sites developed in many patients. It was 71.8% in the local recurrence group when the local recurrence was confirmed. Furthermore, the ratio of patients with radical surgical treatment after local recurrence in the local recurrence group (59.0%) was significantly higher in the no-recurrence group (28.9%). These reasons may affect the results of DFS and OS.

This study has several limitations. First, it is a retrospective, single-institution study. Second, the pathological confirmation of CRLM is limited. However, it is worth noting that despite 38.6% of the CRLM lesions being pathologically diagnosed, all the ablated CRLM were radiologically diagnosed by Gd-enhanced MRI with DWI and dynamic CT with a 1 mm slice. Furthermore, even though pathological confirmations were warranted in all patients, radiological findings (preoperative MRI, CT, and intraoperative ultrasonography) were ascertained in each patient, with or without pathological confirmation. The third limitation is the difference in the MWA device. In this study, a second-generation microwave system was used. Eligible effects of new-generation MWA have been reported [9,11]. The ablation zone formed by overlapping single eradiation with a diameter of 1 cm from the surroundings to the inside differed from the new-generation MWA procedure. However, this study showed sufficient capability for treatment success. Furthermore, this study’s local recurrence rate of 4.1% is comparable to that of 5.4–6.6% after new generation of MWA [9,11]. The fourth limitation is the technical problems. Detailed data on the distance of the puncture from the liver surface or the angle of the puncture were not obtained in this study. It was proposed that a longer distance of puncture, a shallower angle of puncture, and a positional relationship to the intrahepatic vasculature may make MWA difficult; however, it may be difficult to accurately obtain this data. The fifth limitation is the selection bias of the treatment option. Another limitation is the lack of pathologic confirmation of complete ablation with tumor-free margins at the end margins [28]. A Biopsy-proven ablation margin assessment is a useful method for confirming that the ablation margin is tumor-free. However, biopsy specimens from the center and margin of the ablation zone may not reflect tumor necrosis or viability within the entire ablated tissue volume as opposed to the excised surgical specimens. The last limitation is the lack of tumor biological data for rat sarcoma viral oncogene homology (RAS) [20,29] and v-raf murine sarcoma viral oncogene homolog B (BRAF) [29,30]. It was reported that RAS mutations were significantly related to local recurrence after ablation [20].

Furthermore, the development of many kinds of inspection technology may improve ablation margin assessment [15]. The contribution of aiding software systems, 3D imaging reconstruction [31], fusion imaging, and augmented reality are important developments that may contribute to improving the efficacy of the procedure by allowing better planning and more accurate guidance for puncture to obtain the desired trajectory and ablation zones [32]. Furthermore, intraprocedural ablation margin assessment using ammonia perfusion positron emission tomography may contribute to improving the efficacy of the ablation [33].

## 5. Conclusions

This study elucidated the risk factors for local recurrence after MWA for CRLM. In conclusion, MWA is an effective therapeutic option for patients with CRLM who are unfit for liver resection or patients with bilobular multiple CRLM. The outcome of this study may help determine indications for MWA. If local recurrence can be reduced, MWA can be a more effective treatment option for CRLM. Furthermore, the development of inspection technology may help improve the efficacy of the ablation in the future.

## Figures and Tables

**Figure 1 cancers-15-00122-f001:**
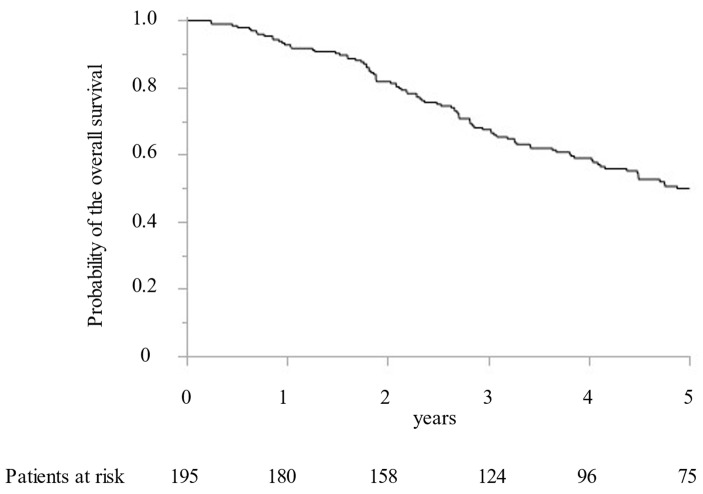
The Kaplan–Meier curve of the overall survival following microwave ablation.

**Figure 2 cancers-15-00122-f002:**
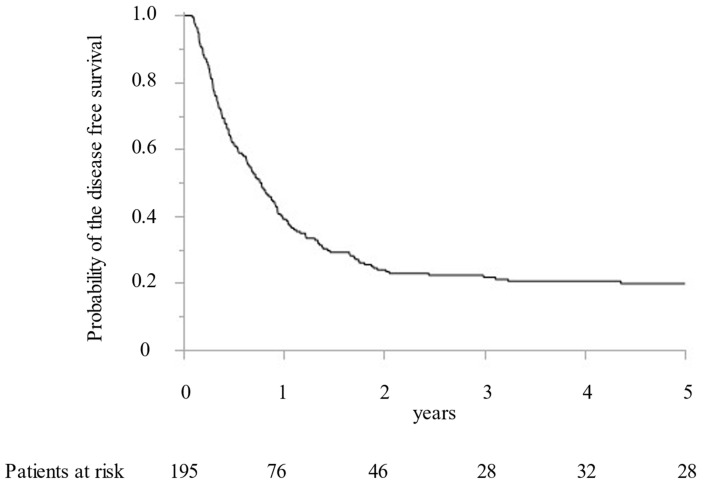
The Kaplan–Meier curve of the disease-free survival following microwave ablation.

**Figure 3 cancers-15-00122-f003:**
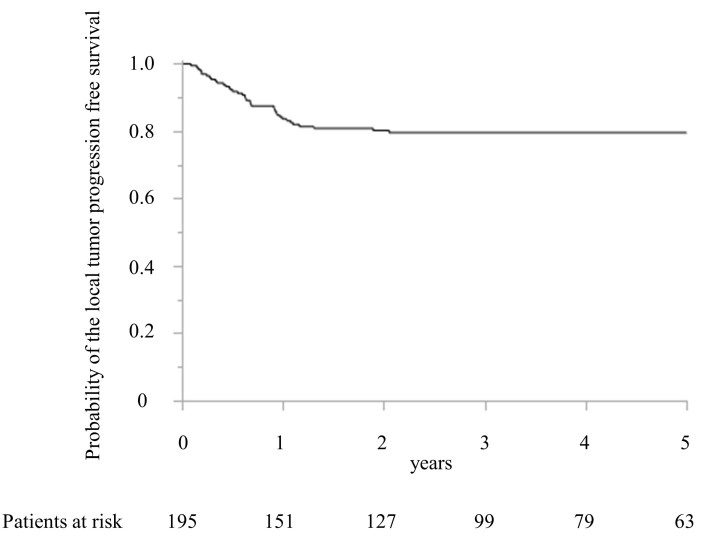
Kaplan–Meier curve of the local tumor progression-free survival after microwave ablation.

**Figure 4 cancers-15-00122-f004:**
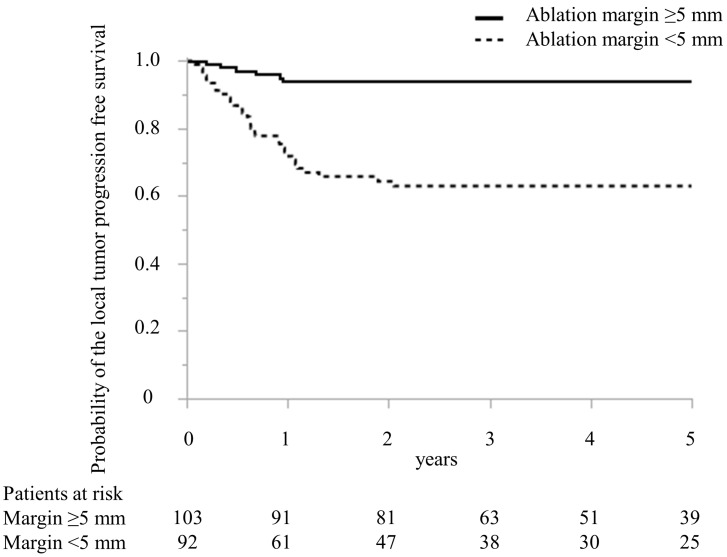
Comparison of Kaplan–Meier curve of the local tumor progression-free survival after microwave ablation between patients with ablation margin >5 mm and ≤5 mm treated with microwave ablation.

**Table 1 cancers-15-00122-t001:** Comparison of baseline characteristics of CRLM in patients with and without local recurrence following treatment with MWA.

Variables		Local Recurrence (*n* = 39)	No Local Recurrence (*n* = 156)	*p*-Value
Age	(median, range)	62 (42–81)	66.5 (35–90)	0.12
Sex	men/women	23/16	102/54	0.46
Timing of CRLM	synchronous/metachronous	27/12	74/82	0.01
Largest diameter of CRLM with MWA or HR	median (range) (mm)	24.8 (13.9–80)	23.3 (3–110)	0.10
Largest diameter of CRLM with MWA	median (range) (mm)	24 (11–44.1)	19.0 (4.8–42)	<0.01
Number of CRLM with MWA and HR	median (range)	6 (1–34)	4 (1–48)	0.051
solitary/multiple		6/33	33/123	0.42
<5/≥5		15/24	87/69	0.053
Number of CRLM with MWA	median (range)	5 (1–34)	3 (1–45)	0.01
solitary/multiple		7/32	49/107	0.09
<5/≥5		18/21	103/53	0.02
Operative procedure				
MWA		24	85	0.43
MWA + HR		15	71	
CEA (ng/mL)	median (range)	10.1 (1.4–1640)	13.0 (0.7–486)	0.94
CA19-9 (U/mL)	median (range)	27 (1–745)	20 (1–120,000)	0.67
Clinical Risk Score	0–2	16	74	0.59
	3–5	23	84	
Preoperative chemotherapy		21 (53.9%)	50 (32.1%)	0.01
Adjuvant chemotherapy		18 (47.4%)	50 (32.1%)	0.041
Primary site				
T stage	1/2/3/4	0/3/31/5	3/7/125/21	0.43
N status	Positive	25	104	0.46
Location of primary site	right/left	10/29	42/114	0.87
Minimal ablation margin	<5 mm, ≥5 mm	33/6	59/97	<0.01
Recurrence in 2 years after surgery		39 (100%)	110 (70.5%)	<0.01
Intrahepatic recurrence		28 (71.8%)	75 (49.4%)	0.01
Extrahepatic recurrence		10 (25.6%)	62 (39.7%)	0.10
RAS status	wild/mutated/not evaluated	14/10/15	46/26/84	0.63

CRLM, colorectal liver metastases; MWA, microwave ablation; HR, hepatic resection; CEA, carcinoembryonic antigen; CA19-9, carbohydrate antigen 19-9.

**Table 2 cancers-15-00122-t002:** Univariate and multivariate analysis of predictive factors for local recurrence in CRLM patients following treatment with MWA.

Variables		Univariate			Multivariate		
		OR	95% CI	*p*-Value	OR	95% CI	*p*-Value
Age	≤62	2.07	(1.02–4.23)	0.045	1.35	(0.57–3.22)	0.50
Sex	Men	0.76	(0.37–1.56)	0.46			
Timing of CRLM	Synchronous	2.50	(1.18–5.27)	0.02	1.62	(0.63–4.14)	0.31
Largest diameter of CRLM with MWA or HR	>23 mm	1.79	(0.86–3.69)	0.12			
Largest diameter of CRLM with MWA	>20 mm	3.38	(1.54–7.42)	<0.01	4.79	(1.83–12.50)	<0.01
Number of CRLM with MWA and HR	≥5	2.02	(0.98–4.14)	0.056	0.58	(0.10–3.39)	0.54
Number of CRLM with MWA	≥5	2.27	(1.11–4.62)	0.02	1.35	(0.24–7.72)	0.74
Operative procedure	MWA + Hr	0.75	(0.36–1.53)	0.43			
Clinical Risk Score	>2	1.23	(0.60–2.51))	0.56			
CEA	≥5 ng/ml	0.88	(0.40–1.90)	0.74			
CA19-9	≥37 U/ml	1.63	(0.79–3.37)	0.19			
Preoperative chemotherapy	Yes	2.47	(1.21–5.05)	0.01	2.61	(0.97–7.02)	0.06
Adjuvant chemotherapy	Yes	2.10	(1.02–4.31)	0.043	2.07	(0.84–5.09)	0.11
minimal ablation margin	<5 mm	9.04	(3.57–22.87)	<0.01	8.32	(2.99–23.13)	<0.01
RAS status	Mutated	1.26	(0.49–3.25)	0.63			

OR, odds ratio; CI, confidential interval.

**Table 3 cancers-15-00122-t003:** Characteristics of tumorous lesions treated with MWA.

Variables	Total						Local Recurrence						
	Lesions	Size (mm)	Range	Ablation Margin (mm)	HV/IVC	Glisson	Lesions	(% of Total)	Size (mm)	Range	Ablation Margin (mm)	HV/ IVC	Glisson
				(0/0.1–5/5–9.10/10 </NE)							(0/0.1–5/5–9.10/10 </NE)		
Total	1066	10	(2.7–44.1)	42/65/874/19/66	42	15	44	(4.1%)	23.0	(10–44.1)	18/20/5/0	18	7
Segment 1	44	12.5	(3.0–37.0)	7/7/28/0/2	7	1	2	(4.5%)	18.8	(13.0–24.5)	1/1/0/0	1	1
2	101	8.5	(2.9–32.9)	1/2/85/0/13	1	2	1	(1.0%)	25	(25.0–25.0)	0/1/0/0	0	1
3	91	8.7	(2.7–28.5)	0/0/80/3/8	0	1	0	(0%)					
4	163	9.2	(2.9–31.6)	6/13/134/0/10	6	2	3	(1.8%)	20	(15.0–24.0)	1/1/1/0	1	1
5	129	9	(3.0–41.3)	0/7/109/6/7	0	1	2	(1.6%)	11.3	(11.0–11.6)	0/2/0/0	2	0
6	151	10.3	(3.5–42.0)	13/19/121/2/7	0	2	2	(1.3%)	24.8	(16.9–32.6)	0/2/0/0	0	1
7	162	10.3	(3.1–41.8)	13/19/121/2/7	13	1	22	(13.6%)	20.5	(10.0–41.8)	9/10/3/0	9	1
8	225	10.1	(2.7–44.1)	15/5/183/6/16	15	5	12	(5.3%)	26.7	(14.4–44.1)	7/3/1/0	7	2

HV, hepatic vein; IVC, inferior vena cava; NE, not evaluated.

**Table 4 cancers-15-00122-t004:** Predictors for local recurrence after MWA in tumorous lesions ablated with MWA.

Variables	Univariate			Multivariate		
	OR	95% CI	*p*-Value	OR	95% CI	*p*-Value
Tumor size > 20 mm	9.98	(5.29–18.81)	<0.01	3.13	(1.05–9.32)	0.04
Posterosuperior segment (1, 7, and 8)	6.93	(3.18–15.09)	<0.01	1.56	(0.49–4.98)	0.45
Posterosuperior segment and > 15 mm	17.2	(8.82–33.53)	<0.01	14.33	(4.56–45.06)	<0.01
Ablation margin (<5 mm)	49.99	(19.34–129.23)	<0.01	45.76	(14.26–145.85)	<0.01
Direct contact with HV/IVC	29.97	(14.46–62.11)	<0.01	1.10	(0.36–3.35)	0.86
Proximity to Glissonian pedicle	24.67	(8.48–71.74)	<0.01	9.08	(2.19–38.58)	<0.01

**Table 5 cancers-15-00122-t005:** Univariate and multivariate analysis of the prognostic factors for disease-free survival in patients with CRLM treated with MWA.

Variables		Univariate			Multivariate		
		HR	95% CI	*p*-Value	HR	95% CI	*p*-Value
Age	>62	0.81	(0.58–1.12)	0.20			
Sex	Men	1.20	(0.59–1.16)	0.28			
Timing of CRLM	Synchronous	1.54	(1.12–2.12)	<0.01	1.09	(0.75–1.61)	0.64
Largest diameter of CRLM with MWA or HR	>5 cm	1.32	(0.80–2.06)	0.26			
Largest diameter of CRLM with MWA	>20 mm	1.42	(1.04–1.96)	0.03	1.28	(0.87–1.90)	0.21
Number of CRLM with WMA and HR							
solitary/multiple	Multiple	2.44	(1.58–3.94)	<0.01	0.92	(0.38–2.20)	0.85
<5/≥5	≥5	2.63	(1.90–3.66)	<0.01	1.5	(0.78–2.88)	0.22
Number of CRLM with MWA							
solitary/ multiple	Multiple	2.67	(1.82–4.01)	<0.01	1.92	(0.92–4.02)	0.08
<5/≥5	≥ 5	2.66	(1.92–3.69)	<0.01	1.34	(0.74–2.44)	0.34
Operative procedure	MCN	0.66	(0.48–0.90)	0.01	0.84	(0.56–1.26)	0.40
CEA	>5 ng/mL	1.55	(1.10–2.23)	0.01	1.27	(0.83–1.94)	0.26
CA19-9	>37 U/mL	1.57	(1.12–2.18)	0.01	1.46	(1.01–2.13)	0.048
Clinical Risk Score	> 2	1.53	(1.11–2.11)	0.01	0.95	(0.64–1.42)	0.81
Preoperative chemotherapy	Yes	1.59	(1.15–2.19)	0.01	1.04	(0.69–1.56)	0.86
Adjuvant chemotherapy	Yes	0.93	(0.67–1.28)	0.64			
Primary site							
T-stage	4	1.53	(0.94–2.37)	0.09			
N-status	Positive	1.21	(0.86–1.72)	0.28			
Right/left side	Right	0.88	(0.60–1.25)	0.48			
Minimal ablation margin	<5 mm	1.74	(1.27–2.39)	<0.01	1.27	(0.87–1.86)	0.21
Local recurrence	Yes	1.85	(1.26–2.66)	<0.01	1.04	(0.66–1.63)	0.86

**Table 6 cancers-15-00122-t006:** Univariate and multivariate analysis of prognostic factors for overall survival in patients with CRLM treated with MWA.

Variables		Univariate			Multivariate		
		HR	95% CI	*p*-Value	HR	95% CI	*p*-Value
Age	>62	1.15	(0.78–1.73)	0.48			
Sex	Men	1.21	(0.82–1.81)	0.34			
Timing of CRLM	synchronous	1.34	(0.92–1.96)	0.13			
Largest diameter of CRLM with MWA or HR	>5 cm	1.71	(0.97–2.83)	0.06			
Largest diameter of CRLM with MWA	>20 mm	1.24	(0.86–1.82)	0.25			
Number of CRLM with MWA and HR							
solitary/multiple	multiple	1.89	(1.14–3.33)	0.01	1.56	(0.50–4.84)	0.44
<5/≥5	≥5	1.64	(1.13–2.40)	<0.01	1.41	(0.67–2.99)	0.37
Number of CRLM with MWA							
solitary/ multiple	multiple	1.69	(1.09–2.71)	0.02	1.50	(0.62–3.59)	0.36
<5/≥5	≥5	1.46	(1.00–2.11)	0.050	0.81	(0.42–1.58)	0.54
Operative procedure	MWA	0.68	(0.46–0.98)	0.04	1.01	(0.60–1.69)	0.97
CEA	>5 ng/mL	2.01	(1.30–3.23)	<0.01	1.51	(0.92–2.49)	0.10
CA19-9	>37 U/mL	1.82	(1.23–2.66)	0.01	1.75	(1.13–2.72)	0.01
Clinical Risk Score	> 2	1.74	(1.19–2.56)	0.01	0.79	(0.38–1.64)	0.53
Preoperative chemotherapy	Yes	1.42	(0.97–2.08)	0.07			
Adjuvant chemotherapy	Yes	0.88	(0.59–1.30)	0.54			
Primary site							
T stage	4	1.77	(1.00–2.93)	0.051			
N status	positive	1.60	(1.05–2.52)	0.03	1.91	(0.97–3.76)	0.06
Right/left side	right	1.22	(0.79–1.84)	0.36			
Minimal ablation margin	<5 mm	1.07	(0.74–1.55)	0.73			
Local recurrence	Yes	1.51	(0.97–2.28)	0.07			

## Data Availability

The data presented in this study is available within the article.

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
