# Peer review of "Predictive Factors for Local Recurrence after Intraoperative Microwave Ablation for Colorectal Liver Metastases"

_cancers, 2022, doi:10.3390/cancers15010122_

Round 1

Reviewer 1 Report

The authors have evaluated 197 patients afert MWA of colorectal liver metastastis. Regarding the novelty of the reported data, similar results have already been published, see below. What is the novelty of this study?

Ausania F. Microwave ablation of colorectal liver metastases: Impact of a 10-mm safety margin on local recurrence in a tertiary care hospital.Ann Hepatobiliary Pancreat Surg. 2021 Aug 31;25(3):366-370

Abreu de Carvalho LF. Local control of hepatocellular carcinoma and colorectal liver metastases after surgical microwave ablation without concomitant hepatectomy. Langenbecks Arch Surg. 2021 Dec;406(8):2749-2757. 

Author Response

Dear Reviewer 1,

We thank you for your thoughtful suggestions and insights. The manuscript has been rechecked and the necessary changes have been made in accordance with the reviewers’ suggestions. The responses to all comments have been prepared and attached given below.

Q1)The authors have evaluated 197 patients afert MWA of colorectal liver metastastis. Regarding the novelty of the reported data, similar results have already been published, see below. What is the novelty of this study?

A1)Thank you for your kind comment. The novelty of this study is that the anatomical location of the posterosuperior segment with tumor size >15 mm was identified as one of the independent predictors for local recurrence following intraoperative MWA. Furthermore, to best of our knowledge, this is the cohort study for the largest number of CRLM lesions evaluating local recurrence following treatment with MWA for CRLM.

Author Response

Dear Reviewer 2,

We thank you for your thoughtful suggestions and insights. The manuscript has been rechecked and the necessary changes have been made in accordance with the reviewers’ suggestions. The responses to all comments have been prepared and attached given below.

Simple Summary

Q1) A mention to why surgical MWA was offered instead of wedge or percutaneous ablation is desired.

A1) Thank you for your kind comment. Following your suggestion, the simple summary was modified as follows.

Based on the concept of non-touch isolation to the tumor, our unique MWA procedure showed that a spherical ablation area with intentional ablation margin of 5‒10 mm was formed by overlapping ablation area (40‒85 W for 30 sec, 1 cm diameter/session) from the tumor surroundings to the inside. (Page 1, lines 17 to 20)

This is why the open approach was considered optimal for ablation at our institution, not the percutaneous approach. (Page 5, lines 160 to 161)

Q2) The study ignores the most important factor of local tumor control and the entire and large body of work on the impact of margins on local tumor control after thermal ablation. Major revision after considering these papers is needed.

A2) Thank you for your kind advice. We have reanalyzed our data using the ablation margin factor, which is the most important predictor.

Q3) This study claims to be the largest intraoperative series of microwave ablation. Maybe the number of tumors reported is one of the largest however the patient number is not.  A key limitation is a proof that the entire tumor number treated was indeed metastatic malignant deposits from CRC. Please see references suggested at the end of this review and revise accordingly.

A3) Thank you for your kind comment. As you have mentioned, the number of tumors is one of the largest; however, the number of patients is small. The manuscript was revised as follows,

To the best of our knowledge, this is the cohort study for the largest number of CRLM lesions evaluating local recurrence following treatment with MWA for CRLM. (Page 15, Lines 317 to 319) (in the Discussion section)

Abstract

Q4) The number of MWA offered at time of resection vs MWA as a stand-alone treatment need be indicated.

A4) We appreciate your essential advice. The abstract was revised as follows,

Data from 195 patients with 1392 CRLM lesions treated with intraoperative MWA (2450 MHz) with or without hepatectomy from January 2005 to December 2019 were retrospectively reviewed and analyzed using logistic regression (Page 1, lines 27 to 30).

The Materials and Methods section was also revised as follows:

From January 2005 to December 2019, 284 patients with initial CRLM underwent surgical treatment (MWA, HR, or MWA+HR). Among these, 199 underwent MWA with or without HR. (Page 4, Line 89 to 92) (in the materials and methods section)

Q5) For those patients that MWA was the sole treatment, why was the surgical vs the percutaneous option preferred/offered?

A5) Thank you for your kind comment.

Based on the concept of non-touch isolation to the tumor, our unique MWA procedure showed that a spherical ablation area with intentional ablation margin of 5‒10 mm was formed by overlapping ablation area (40‒85 W for 30 sec, 1 cm diameter/session) from the tumor surroundings to the inside. (Page 1, Line 17 to 20)

Therefore, we choose the open approach for our MWA procedure.

Q6) Is posterior superior segment referring to segment 7?  Clarification in abstract is desired.

A6) Thank you for your comment. The posterosuperior segment was changed to segments 1, 7, and 8 in the abstract. (Page 1, line 34)

Q7) How was confirmation of metastatic CRC made for the 1082 lesions?

A7) Thank you for your kind comment. The manuscript was revised as follows,

The eligible inclusion criteria were as follows: 2) all target CRLM lesions with assessment by contrast-enhanced CT or contrast-enhanced magnetic resonance imaging (MRI) preoperatively; 3) histopathological diagnosis of CRLM by intraoperative biopsy in at least one or more lesions. (Page 4, line 100 to 104) (in the Methods section)

Q8) Were all lesions confirmed as metastases by at least pre-ablation anatomic imaging or were some only seen with ultrasound during surgery? How do we know that all ablated lesions were metastatic CRC?

A8) Thank you for your kind comment. All lesions were confirmed as metastases by radiological examination using contrast-enhanced CT and MRI. It was desired that all lesions were histopathologically confirmed by biopsy. In this study, histopathological confirmation was performed at least one or more lesions by biopsy each patient, but not for all lesions.

Q9) What is the definition of successful MWA and how was this assessed?

A9) Thank you for your comment. The explanation in the manuscript was insufficient. Hence, we have revised the manuscript as follows.

Complete ablation of a CRLM lesion was established as the absence of residual tumor within 1 cm of the ablation site at the first control imaging with contrast-enhanced CT 6 weeks after MWA. (Page 6, lines 169 to 170) (in the Methods section)

Q10) Have the authors reported according to the Guidelines for Ablation reporting standards? This is extremely important. Please follow relevant guidelines suggested in missing references.

A10) Thank you for your comment. Following your advice, the manuscript has been revised and the reference added as follows.

The first post-MWA contrast-enhanced CT was performed within 6 weeks according to the guidelines (ablation guideline by Pujik et al.) for evaluating treatment efficacy. (Page 4, line 105 to 107)

Q11) A prior paper has indicated that perivascular tumor location impacted outcomes for RFA but not for MWA. When data were stratified by margin there was no difference in local tumor control in the two methods. Please review and comment (Shady et al JVIR 2018). This same paper indicated that MWA was not impacted by the heat sink phenomena. Perivascular tumor location impacted the RF but not the MWA ability to provide sustained local tumor control. Please review and comment.

A11) Thank you for your comment. This study was reanalyzed using a factor of minimal ablation margin. As you supposed, perivascular tumor location was not a significant predictor for local recurrence. These findings have been described in the results and the discussion sections.

Introduction

Q12) Ablation in areas that resection cannot achieve R0 should be assessed differently. Please review Ablation dedicated papers and revise accordingly. A modified clinical risk score for ablation has been described in at least 2 prior publication and impacted outcomes. These papers need be reviewed and used in a significant revision that is indicated for this work.

A12) Thank you for your advice. The predictor for local recurrence was reanalyzed including the modified clinical risk score; however, clinical risk score was not significantly related to local recurrence. (Tables 1 and 2)

Q13) When stratified by the ability to create a margin, outcomes of RF vs MWA seem to be identical. However, MWA seems to be resistant to the heat sink limitations of RFA. Please review and incorporate data of the CLOCC trial as well as ESMO and NCCN guidelines for the use of thermal ablation for CRC liver metastases.

A13) Thank you for your advice. The findings of the CLOCC trial and the ESMO guideline was quoted in the introduction section, as follows.

Non-surgical alternative treatments have been introduced for patients with unresectable CRLM. Among these, radiofrequency ablation (RFA) is the most frequently used. In a randomized phase II study (CLOCC trial) on patients with unresectable CRLM, RFA significantly affected progression-free and overall survival [4]. RFA is widely used in the surgical and radiological community as a reliable treatment modality for patients with unresectable CRLM [5].

Hepatic resection (HR) remains the gold standard for patients with resectable CRLM. Recently, the European Society of Medical Oncology classified surgical resection and thermal ablation as local ablative treatments included in the treatment algorithm for oligometastatic diseases [6], underlining the importance of a multidisciplinary approach when dealing with patients with oligometastatic disease. (Page 3, line 50 to 61)

Q14) Please review and discuss the value of biopsy-proven complete ablation with margins with relevant references.

A14) Thank you for your kind comment. The manuscript was revised as follows,

The other limitation is the lack of pathologic confirmation of complete ablation with margins free of tumor cells [24] at the end of ablation. Biopsy-proven ablation margin assessment is a useful method for confirming that the ablation margin is tumor-free. (Page 17, line 396 to 399) (in the discussion section)

Materials and Methods.

Q15) It would be useful to stratify outcomes by the clinical risk score categories (as previously modified for thermal ablation) and ablation margins as previously described by several papers.

A15) Thank you for your advice. The predictor for local recurrence was reanalyzed including the modified clinical risk score; however, clinical risk score was not significantly related to local recurrence. (Table 1 and 2)

Q16) The term necrotic need be replaced by the imaging surrogate that is used unless there was pathologic confirmation of necrosis

A16) Thank you for your advice. The term “necrotic” was removed.

Q17) The authors describe that an attempt to create 10 mm margins was made. This need be described in much more detail. How was this measured and confirmed? This is a key limitation of this study.

A17) Thank you for your comment. Our description was insufficient. Therefore, the manuscript has been revised as follows:

Intraoperative ultrasonography was routinely performed in all patients to identify lesions and monitor treatment effects and ablation margin. Furthermore, intraoperative contrast-enhanced ultrasonography was used appropriately. (Page 5, line 151 to 153)

 An overlapping ablation area of 1 cm diameter per single session from the tumor surroundings to the inside formed a spherical ablation area with an intentional ablation margin of 5‒10 mm. (Page 5, line 161 to 163) (in the Methods section)

Q18) Why is there a difference in the eligibility tumor size based on location?

A18) The reason was described in the discussion section as follows,

We appreciate your comments. These findings indicate that smaller-sized lesions have a low risk for local recurrence even if the lesions are located in an anatomical position with difficulty for MWA procedure. However, the condition of lesion size may also affect treatment effectiveness. (Page 16, line 351 to 354)

Q19) Were all ablated tumors confirmed metastases and if so, how?

A19) Thank you for your comments. All ablated tumors were radiologically diagnosed liver metastases preoperatively. One or more lesions were pathologically diagnosed by biopsy in each patient.

2.3 F-U and diagnosis of recurrent CRLM

Q20) NCCN guidelines now suggest metabolic imaging in the follow-up algorithm post-ablation to detect early local recurrence including within the ablation zone: Has this been considered? And if so, is it possible that some recurrences were missed due to the lack of follow-up with metabolic imaging?

A20) Thank you for your comment. As you have mentioned, metabolic imaging such as FDG-PET is useful for detecting early local recurrence; however, we did not use the routine FDG-PET owing to high patient cost. When the local recurrence was suspected by contrast-enhanced CT or MRI, FDG-PET is available depending on the cases.

Statistics

Q21) The ablation margins must be calculated and the minimal ablation margin must be included as a factor in the univariate analysis. If significant (and my guess is that it will be) multivariate analysis will need to be reassessed.

A21) Thank you for your advice. Following your advice, the ablation margins were calculated, and the minimal ablation margin were included as a factor in the analysis. Multivariate analysis showed that the minimal ablation margin was a significant predictor.

Results

Q22) The definitions must comply with reporting standards for ablation. Ablation success need be defined and reported. Guidelines recommend assessment by CT or MRI with contrast within 24 hours from ablation and again 3-8 weeks after ablation in order to declare success. Please review and adjust. 

A22) Thank you for your comment. In this study, ablation success was confirmed by contrast-enhanced CT within 1 and 6 weeks after ablation. CT assessment within 24 h is desired; however, our procedure was performed using the open approach. Hence, we adopted CT assessment within 1 week.

Table 1:

Q23) What is the difference of CRLM size with and without MWA?

A23) Thank you for your comment. The description made misleading. This study investigated patients with CRLM treated with MWA alone and MWA plus hepatic resection. The description has been revised to “Largest diameter of CRLM with MWA or HR” and “Largest diameter of CRLM with MWA.”

Q24) Data need be reassessed after stratifying by margin size. I suggest that at the very minimum the methodology described by Wang et al CVIR 2013, is applied using the first available contrast enhance CT or MRI after ablation. Subsequently ablation zones and groups should be stratified as 0 minimal margin, 1-4.9 mm, 5-9.9 and over 10 mm minimal margin. This stratification is essential to allow any meaningful conclusions.  

A24) Thank you for your advice. This study was reanalyzed using a factor of minimal ablation margin. As you have suggested, ablation margin size was identified as one of the significant predictive factors for local recurrence.

Q25) Was there a margin technically achievable when the tumor abutted the HV or the IVC, at least with regards to the direction contrary to the vessel? Please Review papers regarding thermal ablation in perivascular tumors as those provided in the list of references missing below. 

A25) Thank you for your comment. When the tumor is in direct contact with the hepatic vein, it is difficult to achieve effective margin to the direction of the vessels. The minimal ablation margin for lesions with direct contact with the hepatic vein accounted for a margin size of 0 mm.

Q26) I recommend revisiting multivariate analysis for OS after accounting for margins and stratifying by clinical risk score as it has been done in prior papers.

A26) Thank you for your advice. Multivariate analysis for OS was reassessed using clinical risk score. However, this study included patients with remarkable multiple CRLM; hence, the analysis for OS per patient did not include a factor of ablation margin. Instead, the ablation margins were considered in the analysis per lesions.

Q27) Genetic and molecular markers especially Ras mutation has been shown to impact local tumor control for ablation. Please review relevant papers and comment. Is this information available in this cohort?

A27) Thank you for your comment. Comments for molecular markers such as RAS and BRAF were added in the discussion section. RAS data were included in Tables 1 and 2. The data of RAS status was lacking in approximately half of the patients. In addition, an effective RAS analysis was difficult in this study.

Discussion

Q28) 3D software and immediate assessments with metabolic imaging and  tissue sampling; have been shown to improve or predict local control after thermal ablation. All these methods need be discussed and considered in a completely revised discussion after reviewing the many relevant and missing references.

A28) Thank you for your advice. The discussion section was revised as follows,

Furthermore, the development of many kinds of inspection technology may improve ablation margin assessment [14]. The contribution of aiding software systems, 3D imaging reconstruction [27], fusion imaging, and augmented reality are important developments that may contribute to improving the efficacy of the procedure, allowing better planning and more accurate guidance for puncture to obtain the desired trajectory and ablation zones [28]. Furthermore, intraprocedural ablation margin assessment using ammonia perfusion positron emission tomography may contribute to improving the efficacy of the ablation [29]. (Page 17, lines 404 to 411).

Q29) Will need complete revision after accounting for comments above and the suggested papers below

A29) Thank you for your advice. Following your advice, the manuscript has been revised, and the papers you recommended have been quoted in the manuscript.

Reviewer 3 Report

This is a retrospective study to be aimed to investigate predictive factors of occurrence of local recurrence after microwave ablation therapy for colorectal hepatic metastases.

There were some issues to be clarified in this study.

1, Local recurrence after ablation therapy for liver tumors is generally thought to be viable tumors which grow up in previously ablated tumor. In this study, local recurrence was defined as a recurrent tumor in contact with the ablated area in page 3, line 6-7. How did authors discriminate newly developed tumor close to previously ablated tumor? How much of contact was defined as local recurrence in this study. It seems to be very difficult to discriminate these lesions.

2, There were nothing revealed about other recurrent sites rather than local recurrence. Just local and no local recurrence classification was shown on Table1. There must be non-local intrahepatic recurrence, extra-hepatic recurrence such as lung, bone, primary local etc. These other recurrence sites with or without local recurrence should be clearly shown in this study.

3, How was intrahepatic metastatic tumor of proximity to Glissonian pedicle evaluated in this study?

There are the first branch Glissonian pedicle or the IInd branch or the IIIrd branch ones etc. Which order of Glissonian pedicle did this mean in this study? Authors should clarify about it.

4, Multivariate analysis for predicting RFS and OS showed that local recurrence was not significantly predictive value in this study. Why did authors consider that various predictive factors for local recurrence following microwave ablation might be useful for considering the indication of microwave ablation to hepatic metastases despite of these results? It might not make sense from this logistic consideration.  

Author Response

Reviewer 3

We thank you for your thoughtful suggestions and insights. The manuscript has been rechecked and the necessary changes have been made in accordance with the reviewers’ suggestions. The responses to all comments have been prepared and attached given below.

Q1) Local recurrence after ablation therapy for liver tumors is generally thought to be viable tumors which grow up in previously ablated tumor. In this study, local recurrence was defined as a recurrent tumor in contact with the ablated area in page 3, line 6-7. How did authors discriminate newly developed tumor close to previously ablated tumor? How much of contact was defined as local recurrence in this study. It seems to be very difficult to discriminate these lesions.

A1) Thank you for your comment. We apologize for the insufficient description that caused the confusion.The manuscript was revised as follows.

Complete ablation of a CRLM lesion was established as the absence of residual tumor within 1 cm of the ablation site at the first control imaging with contrast enhanced CT 6 weeks after MWA.

Local recurrence was defined as tumor recurrence at the site of ablation after at least one imaging without evidence for rest tumor. (Page 6, line 169 to 172)

Q2) There were nothing revealed about other recurrent sites rather than local recurrence. Just local and no local recurrence classification was shown on Table1. There must be non-local intrahepatic recurrence, extra-hepatic recurrence such as lung, bone, primary local etc. These other recurrence sites with or without local recurrence should be clearly shown in this study.

A2) Thank you for your comment. According to your advice, the data of non-local intrahepatic recurrence and extrahepatic recurrence were added in Table 1.

Q3) How was intrahepatic metastatic tumor of proximity to Glissonian pedicle evaluated in this study? There are the first branch Glissonian pedicle or the IInd branch or the IIIrd branch ones etc. Which order of Glissonian pedicle did this mean in this study? Authors should clarify about it.

A3) Thank you for your comment. Proximity to the Glissonian pedicle was defined as present when a lesion was located at <5 mm of the first to third the branch of Glissonian pedicle. (Page4, Lines 143 to 145).

Q4) Multivariate analysis for predicting RFS and OS showed that local recurrence was not significantly predictive value in this study. Why did authors consider that various predictive factors for local recurrence following microwave ablation might be useful for considering the indication of microwave ablation to hepatic metastases despite of these results? It might not make sense from this logistic consideration.

A4) Thank you for your comment. The primary endpoint of this study was the predictive factors for local recurrence. However, local recurrence may be related to oncological outcomes. We also considered that local recurrence may be related to RFS or OS; however, this study showed that local recurrence was not related RFS or OS. The absence of local recurrence is desirable for ablation. We believe that it is possible to reduce the local recurrence of MWA in the future by clarifying the predictive risk factor for local recurrence.

Reviewer 4 Report

This study evaluated predictive factors for local recurrence after intraoperative microwave ablation. The presented data are relevant for clinicians performing microwave ablation. Below are several issues that I suggest the authors address.

1) The MWA system used is different from MWA systems used in other countries. The systems I am mostly familiar with perform MWA for several minutes, and create ablations in the range of 3-4 cm in diameter after a single application. This is different from the system used in this study, where a single ablation was performed for 30 seconds, and was 1 cm in diameter. It would be helpful to briefly mention these differences in MWA systems in the introduction. Also, some discussion should be added on how the results are applicable to other MWA systems in the discussion section.

2) Please add specifics (power, ablation time, ablation size) in the abstract.

3) How was completeness of ablation confirmed. It is stated that ultrasound imaging was used, but to my knowledge ultrasound alone cannot visualize the ablation zone without contrast.

4) Based on the data, progression-free survival (PFS) and overall survival (OS) was also evaluated. Please add plots of PFS and OS either in Fig. 1, or as additional figures.

5) Add graphs showing local recurrence for relevant subgroups, e.g. for patients with and without preoperative chemotherapy.

6) What kind of chemotherapy was used - was it the same in all patients?

7) change 'surgical' to 'intraoperative' in title

8) Add in discussion section a brief discussion on the impact of local recurrence on PFS and OS.

Author Response

Dear Reviewer 4

We thank you for your thoughtful suggestions and insights. The manuscript has been rechecked and the necessary changes have been made in accordance with the reviewers’ suggestions. The responses to all comments have been prepared and attached given below.

Q1) The MWA system used is different from MWA systems used in other countries. The systems I am mostly familiar with perform MWA for several minutes, and create ablations in the range of 3-4 cm in diameter after a single application. This is different from the system used in this study, where a single ablation was performed for 30 seconds, and was 1 cm in diameter. It would be helpful to briefly mention these differences in MWA systems in the introduction. Also, some discussion should be added on how the results are applicable to other MWA systems in the discussion section.

A1) Thank you for your comment.

The description of our unique MWA procedure was added in the simple summary and methods section as follows: “Based on the concept of non-touch isolation to the tumor, our unique MWA procedure showed that a spherical ablation area with intentional ablation margin of 5‒10 mm was formed by overlapping ablation area (40‒85 W for 30 sec, 1 cm diameter/session) from the tumor surroundings to the inside.” (Page 1, lines 17 to 20).

Furthermore, comment for our unique MWA procedure was added in the discussion section as follows.

“Our unique MWA procedure showed that the ablation zone formed by overlapping single eradiation with a diameter of 1 cm from the surroundings to the inside differed from the new-generation MWA procedure. However, this study showed sufficient capability for treatment success. Furthermore, this study’s local recurrence rate of 4.1% is comparable to that of 5.4‒6.6% after new generation MWA.” (Page 17, line 386 to 391)”

Q2) Please add specifics (power, ablation time, ablation size) in the abstract.

A2) Thank you for your comment. Specifics (power, ablation time, and ablation size) have been added in the simple summary because of word-count limits.

Q3) How was completeness of ablation confirmed. It is stated that ultrasound imaging was used, but to my knowledge ultrasound alone cannot visualize the ablation zone without contrast.

A3) Thank you for your comment. Based on our experiences, we believe that the ablation zone of MWA can be visualized by intraoperative ultrasonography in many cases. However, contrast-enhanced ultrasonography was also used appropriately in this study.

Q4) Based on the data, progression-free survival (PFS) and overall survival (OS) was also evaluated. Please add plots of PFS and OS either in Fig. 1, or as additional figures.

A4) Thank you for your advice. Plots of OS and PFS have been added in Figures 1 and 2, respectively.

Q5) Add graphs showing local recurrence for relevant subgroups, e.g. for patients with and without preoperative chemotherapy.

A5) Thank you for your kind advice. The comparison of the Kaplan‒Meier curve of the local tumor progression-free survival after microwave ablation between patients with a tumor size > 20 mm and ≤ 20 mm treated with microwave ablation has been added in figure 4.

Q6) What kind of chemotherapy was used - was it the same in all patients?

A6) Thank you for your comment. The chemotherapy data has been added in the results section, as follows.

The regimen of preoperative chemotherapy in the local recurrence and no local recurrence groups was oxaliplatin-based doublet in four and four patients, irinotecan-based doublet in 0 and one, cisplatin-based doublet in three and three, bevacizumab and oxaliplatin-based doublet in eight and 27, bevacizumab and irinotecan-based doublet in two and three, epidermal growth factor receptor inhibitor and irinotecan-based doublet in four and eight, and bevacizumab and triplet in 0 and five patients, respectively (p=0.33). (Page 10, lines 254 to 260.

Q7) change 'surgical' to 'intraoperative' in title

A7) Thank you for your comment. Following your advice, “surgical microwave ablation” has been changed to “intraoperative microwave ablation” in the title.

Q8) Add in discussion section a brief discussion on the impact of local recurrence on PFS and OS.

A8) Thank you for your comment. Following your advice, the comment of the impact of local recurrence on DFS and OS has been added in the discussion section, as follows.

Since local recurrence was not significantly related to DFS or OS in this study, almost all patients exhibited multiple CRLM in the local recurrence group, and intrahepatic new recurrence in other liver sites developed in many patients. It was 71.8% in the local recurrence group when the local recurrence was confirmed. Furthermore, the ratio of patients with radical surgical treatment after local recurrence in the local recurrence group (59.0%) was significantly higher in the no-recurrence group (28.9%) (data not shown). These reasons may affect the results of DFS and OS (Page 17, lines 376 to 382).

Round 2

Reviewer 1 Report

The paper has been thoroughly restructured.

Author Response

Dear Reviewer 1,

We thank you for your thoughtful suggestions and insights. The manuscript has been rechecked and the necessary changes have been made in accordance with the reviewers’ suggestions. The responses to all comments have been prepared and attached given below.

Reviewer’s comment

The paper has been thoroughly restructured.

To reviewer 1

We thank you for your review. The manuscript was edited with English language proofreading by Editage (www.editage.com).

Author Response

Dear Reviewer 2,

We thank you for your thoughtful suggestions and insights. The manuscript has been rechecked and the necessary changes have been made in accordance with the reviewers’ suggestions. The responses to all comments have been prepared and attached given below.

Predictive factors for local recurrence after intraoperative mi- 2 crowave ablation for colorectal liver metastases

Simple Summary: 

Q1) The reference of surgery as the gold standard does not add anything to this paper and summary. Instead please adhere to the recommendations for ablation both in the ESMO and NCCN Guidelines for CRC liver metastases, principles of surgery, indicating that Ablation is recommended as a stand alone treatment or in combination with surgery as long as all visible disease is eradicated. The mention of specific technique here is not useful and rather confusing since the author has not yet read the methods. The mention of the margin as the most important factors is the most important message of the paper.

Response: Thank you for your helpful comment. According to your suggestion, we have revised the simple summary and reworded the statement “Hepatectomy is the gold-standard treatment for colorectal liver metastases” as “The European Society of Medical Oncology and National Comprehensive Cancer Network guidelines indicate that ablation is recommended as a stand-alone treatment or in combination with resection for colorectal liver metastases (CRLM) as long as all visible tumors are eradicated.” Furthermore, we emphasized that the ablation margin was the most powerful predictor in the simple summary. (Lines 9–18)

Abstract

Q2) Is the factor > 15 mm referring to tumor size?

Response: Thank you for pointing this out. In the simple summary, we described the factor of segments 1, 7, and 8 separately, and we have explained that the tumor size was >15 mm. (Lines 17–18)

Q3) There still remains the need to provide convincing evidence regarding the malignant nature of the ablated tumor. How was this confirmed?

Response: We appreciate your valuable comment. We have revised the statement on lines 20–22 to “…CRLM lesions, who were preoperatively diagnosed by gadolinium-enhanced MRI with diffusion-weighted imaging and dynamic CT.” Further, we gave more convincing evidence regarding the malignant nature of the ablated tumor in the Methods and the Results section, as follows:

Line 127-131: “All targeted CRLM were preoperatively diagnosed by gadolinium (Gd)-enhanced MRI with diffusion-weighted imaging (DWI) (from 2005 to 2007) or Gd-ethoxybenzyl-diethylenetriamine pentaacetic acid (Gd-EOB-DTPA) enhanced MRI with DWI and hepatobiliary phase image (from 2008). In addition, dynamic CT scans with a 1 mm slice were also preoperatively examined for diagnosis of CRLM, and all targeted CRLM were reconfirmed as solid tumors by intraoperative ultrasonography.”

Lines 194-200: “Preoperative radiological examination confirmed that all targeted CRLM showed a hypovascular tumor with ringed enhancement by Gd-enhanced MRI and dynamic CT, high intensity with T2-weighted imaging and Gd-enhanced MRI with DWI, and low intensity with hepatobiliary phase image by Gd-EOB-DTPA enhanced MRI (from 2008). Furthermore, all targeted CRLM were reconfirmed as solid tumors by intraoperative ultrasonography. Finally, 411 lesions (38.6%) were histopathologically diagnosed as CRLM by intraoperative biopsy after MWA.”

Furthermore, the limitation of the study was revised as follows because not all ablated lesions were pathologically confirmed:

Lines 362–367: “Second, the pathological confirmation of CRLM is limited. However, it is worth noting that despite 38.6% of the CRLM lesions were pathologically diagnosed, all the ablated CRLM were radiologically diagnosed by Gd-enhanced MRI with DWI and dynamic CT with a 1 mm slice. Further, even though pathological confirmations were warranted in all patients, radiological findings (preoperative MRI, CT, and intraoperative ultrasonography) with or without pathological confirmation were ascertained in each patient.”

Introduction

Q4) There needs to be a revision and reflect the recommendation that “Ablation can be used alone or in combination with surgery as long as all visible disease can be eradicated” (NCCN 2022. Principles of Surgery). In the current form the introduction implies that ablation is only indicated for non-resectable disease, while often patients undergoing ablation maybe poor surgical candidates due to comorbidities, or ablated for recurrences after hepatectomy, it is not accurate to give the message that the indication for  ablation is limited to non-resectable disease.

Response: Thank you for your insightful comment. We have revised the introduction as follows:

“Recently, the National Comprehensive Cancer Network (NCCN) [6] and the European Society of Medical Oncology (ESMO) [7] guidelines indicate that ablation is recommended as a stand-alone treatment or in combination with hepatic resection (HR) for CRLM as long as all visible tumors are eradicated.” (Lines 46–49)

Q5) Again instead of repeating that surgery is the gold standard I would stick withESMO and NCCN recommednations specific to ablation.

Response: We appreciate your valuable suggestion. Following your suggestion, the statement, “Hepatic resection (HR) remains the gold standard for patients with resectable CRLM,” was removed, and ESMO and NCCN recommendations specific to ablation were added. (Lines 46–49)

Q6) The most significant message of this work is the identification of the ablation margin as the most important factor for local control in a surgical series of thermal ablation.

Response: Thank you for your insightful comment. We have revised the introduction as follows:

“Recently, ablation margin size has been considered the most important factor for LR after ablation [20-22].” (Lines 63–65)

“Therefore, this study aimed to retrospectively review patients with CRLM treated with surgical MWA in order to identify the predictive factors for LR after intraoperative MWA for CRLM and investigate the relationship between ablation margin and LR.” (Lines 70-72)

Methods

Q7) There is confusing information regarding the time of the first post ablation scan to assess treatment efficacy.

Response: We appreciate your helpful comment. Following your comment, to avoid misunderstanding, we revised the statement as follows:

“CT evidence of effective ablation: All ablations would have been essentially assessed by 6 weeks after MWA contrast-enhanced CT as the first post-assessment according to the guidelines [24] for evaluating the treatment efficacy.” (Lines 91–94)

Q8) When and under what conditions was US contrast used?

Response: Thank you for your comment. We have revised the Methods section as follows:

“…intraoperative contrast-enhanced ultrasonography was performed when ablation was used in treating small-sized CRLM, multiple CRLM, or CRLM lesions expressed unclearly by intraoperative ultrasonography.” (Lines 112–114)

Q9) The comment that the open approach is better than percutaneous is not supported by any data and should be eliminated unless specific data comparing the 2 metods are available.

Response: Thank you for your valuable comment. As you pointed out, the superiority of our open approach over percutaneous was not supported by any specific data; therefore, we have removed the statement.

Q10) A non-touch technique has been described and is also commonly practiced with percutaneous ablation.

Response: We appreciate your insightful comment. Since the unique MWA procedure with an open approach was based on our idea and can not be supported by any specific data, we have deleted the statement “Based on the concept of non-touch isolation to the tumor” from the Methods section.

Q11) Why is larger tumor size allowed only in subcapsular location? What are the data supporting this approach?

Response: Thank you for your comment. Since this was also based on our idea and can not be supported by any specific data, the statement “At our institution, the optimal inclusion criteria for MWA are a CRLM diameter under 3 cm when the CRLM is located on the liver surface and under 2 cm when the CRLM is located deep in the liver” was removed from the Methods section.

Definitions.

Q12) Paper 20 does not describe the modified Clinical risk score for ablation. This is described in Shady et al Radiology 2016 .

Response: Thank you for pointing this out. We apologize for the mistake. The reference has been changed to “Shady et al. Radiology 2016.”

Q13) Please reference the specific “Shady paper” mentioned as a method to measure the margin.

Response: Thank you for your comment. We have referenced Shady's paper. In addition, Wang's paper was added.

Q14) Line 150: Were abaltions offered on vanished/disappeared tumors too? And if so, how were this localized?

Response: We appreciate your comment. We are sorry our former explanation was insufficient. Therefore, we have revised the manuscript as follows:

“Although lesions were detected by intraoperative ultrasonography and the first preoperative contrast-enhanced CT scans before preoperative chemotherapy, when an unclear lesion showed at the last preoperative contrast-enhanced CT scans after preoperative chemotherapy, the minimal margin was regarded as not being evaluated.” (Lines 155–160)

Results

Q15) Still needed is a convincing evidence that all ablated tumors were malignant.

Response: Thank you for your comment. We added more convincing evidence in the Results section as follows:

“Preoperative radiological examinations confirmed that all targeted CRLM showed a hypovascular tumor with ringed enhancement by Gd-enhanced MRI and dynamic CT, high intensity with T2-weighted imaging and DWI, and low intensity with hepatobiliary phase imaging by Gd-EOB-DTPA enhanced MRI (from 2008). Moreover, all targeted CRLM were reconfirmed as solid tumors by intraoperative ultrasonography. Finally, 411 lesions (38.6%) were histopathologically diagnosed as CRLM by intraoperative biopsy after MWA.” (Lines 193–200)

Q16) It is desired to evaluate all significant factors in multivariate analysis (MVA). In particular It is very important to see the relation of margins and tumor size when assessed in MVA

Response: Thank you for your comment. Following your suggestion, multivariate analysis was reanalyzed, including the factors of margins and tumor size. Please see Tables 2 and 4.

Q17) I cannot find the ablation margins in the UV or MVA in disease free survival and Overall survival. Were they assessed at all?

Response: We appreciate your helpful comment. According to your suggestion, the multivariate analyses for DFS and OS were reanalyzed, including the factors of ablation margins. Please see Tables 5 and 6.

Discussion.

Q18) A prior paper indicated that while ablation margins over 10 mm were related with optimal tumor control, they were also associated with increased risk for biliary complications. Please discuss (Kurilova et al, Clinical Colorectal cancer 2021).

Response: Thank you for your helpful comment. We have explained this in the discussion section (Lines 303-309) as follows:

“A previous study [27] reported that the ablation margin ˃ 10 mm offered the best tumor control. Consistent with the above-mentioned study, this study also showed no local recurrence in the lesions with an ablation margin ˃ 10 mm. However, an ablation margin ˃ 10 mm increased the risk of biliary complications after ablation in patients with hepatic arterial infusion chemotherapy. Therefore, the relationship between biliary complications after MWA and ablation margin in patients with preoperative systemic chemotherapy needs to be clarified in the future.”

REFERENCES

We thank you so much for your kind advice. Benson paper and Kurilova paper was referred.

Benson AB et al: Colon Cancer, Version 2.2021, NCCN Clinical Practice Guidelines in Oncology. J Natl Compr Canc Netw. 2021 Mar 2;19(3):329-359. doi: 10.6004/jnccn.2021.0012. PMID: 33724754.

Kurilova I. et al: Factors Associated With Local Tumor Control and Complications After Thermal Ablation of Colorectal Cancer Liver Metastases: A 15-year Retrospective Cohort Study. Clin Colorectal Cancer. 2021 Jun;20(2):e82-e95. doi: 10.1016/j.clcc.2020.09.005. Epub 2020 Oct 24. PMID: 33246789.
